# Enhanced T-Cell Priming and Improved Anti-Tumor Immunity through Lymphatic Delivery of Checkpoint Blockade Immunotherapy

**DOI:** 10.3390/cancers14071823

**Published:** 2022-04-04

**Authors:** Carolina Mantilla-Rojas, Fred C. Velasquez, Janelle E. Morton, Leticia C. Clemente, Edwin R. Parra, Carlos Torres-Cabala, Eva M. Sevick-Muraca

**Affiliations:** 1Center for Molecular Imaging, The Brown Foundation Institute of Molecular Medicine, The University of Texas Health Science Center, Houston, TX 77030, USA; caromanty3000@gmail.com (C.M.-R.); fred.christian.velasquez@uth.tmc.edu (F.C.V.); janelle.e.morton@uth.tmc.edu (J.E.M.); 2Department of Translational Molecular Pathology, The University of Texas M.D. Anderson Cancer Center, Houston, TX 77030, USA; lcampos@mdanderson.org (L.C.C.); erparra@mdanderson.org (E.R.P.); 3Departments of Pathology and Dermatology, The University of Texas M.D. Anderson Cancer Center, Houston, TX 77030, USA; ctcabala@mdanderson.org

**Keywords:** checkpoint blockade immunotherapy, lymphatic delivery, virus like particles, tumor infiltrating lymphocytes, near-infrared fluorescence lymphatic imaging

## Abstract

**Simple Summary:**

While checkpoint blockade immunotherapy results in durable, complete responses in some patients, most cancer types are non-responsive and the majority of patients with cancers known to be responsive do not benefit from these therapies. Herein, we show that antigen directed T-cell priming orchestrated through combined vaccination and checkpoint blockade immunotherapy delivered through the lymphatics can improve anti-tumor responses in otherwise non-immunogenic, syngeneic melanoma models. The work presented herein suggests that by directing immune responses within the regional lymph node environment, more patients could potentially benefit from checkpoint blockade immunotherapies.

**Abstract:**

An infusion of checkpoint blockade immunotherapy (CBI) has revolutionized cancer treatments for some patients, but the majority of patients experience disappointing responses. Because adaptive immune responses are mounted by the concentrated assembly of antigens, immune cells, and mediators in the secluded and protective environment of draining lymph nodes (dLNs), we hypothesize that lymphatic delivery of CBI (αCTLA-4 and αPD-1) to tumor dLNs (tdLNs) improves anti-tumor responses over intravenous (i.v.) administration, and that vaccination against tumor associated antigen (TAA) further enhances these responses. Mono- and combination CBI were administered i.v. or through image-guided intradermal (i.d.) injection to reach tdLNs in vaccinated and unvaccinated animals bearing either primary or orthotopically metastasizing B16F10 melanoma. Vaccination and boost against TAA, Melan-A, was accomplished with virus-like particles (VLP) directed to tdLNs followed by VLP boost after CBI administration. Lymphatic delivery of CBIs reduced primary tumor size and metastatic tumor burden, alleviated the pro-tumorigenic immune environment, and improved survival over systemic administration of CBIs. Animals receiving CBIs lymphatically exhibited significantly enhanced survival over those receiving therapies administered partially or completely through systemic routes. By combining vaccination and CBI for effective T-cell priming in the protected environment of dLNs, anti-tumor responses may be improved.

## 1. Introduction

Successful cancer immunotherapy begins with the presence of tumor-specific cytotoxic T-cells for anti-tumor immunity. Adoptive T-cell therapies use autologous T lymphocytes that are activated and replicated in vitro before being returned to the body to kill tumor cells, while immune checkpoint blockade inhibitor therapies suppress co-inhibitory (or promote co-stimulatory) T-cell signaling in vivo to generate and sustain tumor-specific T-cells. Checkpoint blockade immunotherapy (CBI) offers the broadest form of anti-tumor immunity with antibodies targeting the programmed death 1 (PD1)/PD1 ligand 1 (PD-L1) axis *systemically* administered alone or in combination with antibodies targeting cytotoxic T lymphocyte antigen (CTLA-4), with the expectation that these therapies direct immune responses against tumor-associated antigens (TAA) or tumor neoantigens. Because of anti-CTLA-4 (αCTLA-4) monotherapy is associated with lower response rates and higher rates of Grade 3–4 toxicities than anti-PD-1 (αPD-1) monotherapy [1,2,3], αPD-1 monotherapy has become the preferred immunotherapy in patients with advanced melanoma [4,5]. For those patients unresponsive to monotherapy, the combination of αCTLA-4 and αPD1 therapies has been shown to have complementary activity of up to 50–60% response rates in advanced Stage III or IV melanoma, but disappointingly, they act synergistically to amplify immune-related adverse events (irAEs) and severe toxicity in up to 60% of all patients [2,3,6,7,8]. Sequential monotherapies of αPD-1 followed by αCTLA-4 or the reverse sequence have optimal response rates of 30–40% in advanced metastatic melanoma, but again, with toxicity rates of ~50% [9]. Although confounders of CBI include diet, age, gender, lifestyle, and chronic disorders [10], a focused approach to improve tumor-specific, and alleviate tumor-irrelevant, immune responses could improve response rates. 

Despite the systemic administration of CBI and other immune modifying drugs, immune responses are mounted *regionally* within draining lymph nodes (dLNs) where antigen (Ag) presenting cells (APCs) activate LN-resident, naïve T cells. Following T-cell receptor (TCR) activation in the absence of co-inhibitory signaling, activated T-cells proliferate before leaving the efferent lymphatic vessel for systemic dissemination to the bloodstream to reach the tumor microenvironment (TME). However, T-cell upregulation of CTLA-4 and its binding to B7 can limit activation and subsequent proliferation in LNs. Once successfully activated, T-cells upregulate PD-1, which in early stages of differentiation can exert an inhibitory role [11]. Upon exit from the LN, activated T-cells expressing PD-1 can also be tolerized by PD-L1 expressed on major histocompatibility complex (MHC)-presenting lymphatic endothelial cells (LECs) in efferent lymph vessels and LNs [12,13,14], as well as on tumor, immune, and stromal cells in the TME. Although intravenous (i.v.) delivery of αCTLA-4 and αPD-1/PD-L1 may reach targets in the TME, the critical sites for *initiating anti-tumor immune responses* may be within the lymphatic vessels and LNs [15]. Because LN-resident dendritic cells (DCs) and LECs are capable of presenting MHC I and II molecules, lymphatic vessels and LNs are also sites for *tolerogenic cross presentation* establishing peripheral tolerance [16,17,18]. 

In higher order vertebrates, the lymphatic vasculature system unidirectionally drains lymph and immune cells from individual lymphatic watersheds into the circulatory blood vasculature for systemic immunity [19] (Figure 1). The entry of drugs into LNs can occur through afferent lymphatic vessels and, to a far lesser extent, through high endothelial venules following i.v. delivery [20]. We, therefore, hypothesize that lymphatic delivery of αCTLA-4 and αPD-1 will result in improved anti-tumor responses when compared to i.v. delivery.

Another impediment to CBI is the lack of high mutational burden that renders most tumors non-immunogenic. These cancers are “cold” because TAAs mimic those present in normal tissues and, therefore, escape immune surveillance through central immunological tolerance. However, by delivering a vaccination carrier of TAAs that can elicit both MHC class I and II immunodominant responses in regional dLNs, anti-tumor immune responses may be pharmacologically manipulated regionally. Virus-like particles (VLPs) are noninfectious, multimeric structural particles that elicit pathogen-associated molecular patterns (PAMPs); can be modified to present correctly folded proteins to the immune system [21]; and can induce strong, Ag-specific CD4^+^ and CD8^+^ T-cell responses without the need for adjuvants [22,23]. Because VLPs are used in market-approved vaccination therapies (such as Gardasil produced by Merck and Cervarix, produced by GSK), the combination of CBIs with VLP therapeutic vaccination could be translated for effective treatment of classes of “cold” tumors, without the need for the sequencing required for personalized neoantigen vaccines.

Herein, we demonstrate improved anti-tumor responses when αCTLA-4 and αPD-1 monotherapies, as well as αCTLA-4/αPD-1 combination therapy, are delivered lymphatically, as opposed to systemically, in animal models of otherwise non-immunogenic, B16F10 melanoma. Furthermore, when combined with VLP vaccination against TAA, Melan-A, anti-tumor responses were robust with significant survival benefits. The results highlight the importance of the lymphatic vasculature and dLNs in establishing and shaping successful T-cell priming in CBI and altering TME.

## 2. Materials and Methods

### 2.1. Animals

C57BL/6 female mice were obtained from Charles River, housed five per cage, fed Purina Mills Lab Diet 5053, and maintained at 22° under a 12-h light cycle. For experimental procedures, the mice were anesthetized using 2% isoflurane. Anesthetized mice underwent shaving and depilatory cream (Nair) application before tumor inoculation to ensure precise tumor cell inoculation and tumor growth measurements, and subsequent near-infrared fluorescence lymphatic imaging (NIRF-LI) to ensure CBI delivery to dLNs. 

### 2.2. Reagents and Therapies

The CBIs consisted of anti-mouse CTLA-4 (CD152) clone 9D9 (mouse IgG2b, Cat: BE0164), rat anti-mouse PD-1 (CD279) (rat, IgG2a Kappa, Cat: BE0146) and isotype controls (Cat: BE0086 and BE0089) from BioXCell. For Melan-A VLP, the open reading frame of *Mlana* (*Melan-A*) gene was cloned from B16F10 mouse melanoma cell lines (ATCC, Manassas, VA, USA), introduced into pBacPAK9 plasmid (Clontech, Mountain View, CA, USA, Xat: 631402) and used for producing recombinant baculoviruses expressing *Melan-A* gene by the Protein and Monoclonal Antibody Production Core in Baylor College of Medicine. The chimeric Melan-A VLPs and control VLPs (containing structure protein SIV Gag, Gag-VLPs) were produced and characterized by Q. Yao (Baylor College of Medicine, Houston, TX, USA) according to previously published methods [24] and provided to the project. 

### 2.3. B16F10 Tumor Models and Experimental Treatment Groups

B16F10 melanoma cells from ATCC (CRL-6475™) were cultured in complete Dulbecco’s Modified Eagle’s Medium (ATCC, Cat: 30-2002) supplemented with 10% (*v/v*) Fetal Bovine Serum (VWR, 76236-336) and 2% (*v/v*) Antibiotic-Antimycotic (ThermoFisher, Cat: 15240062), and maintained at 37 °C in a humidified incubator with 5% CO_2_. At 70% confluence, the cells were collected and prepared for in vivo injection. Briefly, 7–8-week-old C57BL/6 female mice were inoculated intradermally (i.d.) on the right hindlimb with 1 × 10^4^ cells B16-F10 cells in a 50 µL solution of 1:1 PBS to Matrigel Matrix, basement membrane (VWR, Cat: 47743-715) on Day 0. Caliper measurement of primary tumor sizes began at 7 days post implantation (d.p.i.), at which time the mice were randomized into treatment groups of (i) αCTLA-4 or αPD-1 monotherapy; and (ii) αCTLA-4 and αPD-1 combination therapy with and without TAA-VLP (Melan-A VLP). The control groups consisted of animals without treatment, isotype IgG2b i.d. with and without TAA-VLP, and non-TAA, control Gag-VLP.

Immunotherapy treatments consisted of 10 mg/kg injected i.v. (tail vein) or i.d. (at right base of tail) at 7, 9, and 11 d.p.i. TAA-VLP (100 μg) was administered i.d. (at right base of tail) at 7 and 14 d.p.i. (Figure 2). The tumor volume (mm^3^) was calculated with the formula: V= 0.5 × (width^2^) × length. The animals were euthanized at 17 d.p.i. for evaluation of their systemic and TME immune status. In another cohort, the influence of route of administration on survival was evaluated by euthanizing animals at 30 d.p.i., when primary tumor size reached 20 mm in any dimension or developed necrosis, or if the animals lost >20% body mass, whichever came first. Peripheral blood mononuclear cells (PBMCs) and the primary tumor were also collected. Samples were stored in 10% formalin for multiplex-immunofluorescence (mIF) or in cell culture media for flow cytometry.

To evaluate the lymphatic delivery of CBIs in a model with an increased prevalence of metastasis through the lymphatics, 7–8-week-old C57BL/6 female mice were inoculated i.d. on the dorsal aspect right hind paw with 1 × 10^4^ cells B16-F10 cells in a 20 µL solution of 1:1 PBS to Matrigel Matrix on Day 0 with CBI dosing identical to Figure 2. Because the primary tumor growth was limited and tumor progression occurred through popliteal and inguinal LN metastasis, *in lieu* of caliper measurements, we collected 100 µL cheek bleeds at 8, 14, and 18 d.p.i. for RNA analyses of mouse *Pmel17*, a protein that is essential for the synthesis of melanin (31). Samples were stored for RNA extraction at −80 °C.

### 2.4. RNA Analysis

RNA was extracted using a QIAamp RNA Blood Mini Kit (Qiagen, Germantown, MD, USA, Cat: 52304) following the manufacturer’s protocol, and the concentrations were determined via UV absorption at 260 nm. Then, 0.5 µg of total RNA was used to synthesize the single stranded cDNA with reverse transcriptase (QuantiTect^®^ Reverse Transcription Kit; Qiagen, Germantown, MD, USA, Cat: 205314) following the manufacture’s protocol. A PCR was performed as recommended by the manufacturer (SsoAdvanced™ Universal SYBR^®^ Green Supermix–Biorad, Hercules, CA, USA, Cat: 1725271). For the detection of *Pmel17* and *Gapdh* mRNAs, 30 cycles of PCR were carried out (95 °C for 1 min, 58 °C for 1 min, and 72 °C for 1 min). The PCR reactions were set up in 96-well plates. All samples were run in triplicate. The analysis was performed on an ABI-7900 (Thermofisher, Waltham, MA, USA). Blood *Pmel17* mRNA expression was reported relative to *Gapdh* mRNA. To evaluate the sensitivity for detecting B16F10 cells in blood, 100 µL aliquots of normal mouse blood were mixed with dilutions of B16F10 melanoma cells (0–10^6^) cells and analyzed for relative expression of *Pmel17* mRNA. 

### 2.5. Confirmation of Lymphatic Delivery via Near-Infrared Fluorescence Imaging (NIRF-LI) and Biodistribution

To confirm the LN delivery of immunotherapies at the time of administration, test vehicles were reconstituted in 645 µM of indocyanine green (ICG) (Akorn, Pharmaceuticals, Lake Forest, IL, USA, 17478-701-02) and, after injection, the animals were illuminated with 785 nm NIRF excitation light and the emission of 830 nm was collected using an intensified, cooled 16 bit charged-coupled device, similar to that used to visualize lymphatic function in clinical studies [25,26]. 

In addition, two cohorts of animals bearing hindlimb tumors were dosed 14–16 d.p.i. with ^89^Zr-labeled αCTLA-4 (0.025 mg, 28–31 μCi in 50μL, i.d. or i.v.) and euthanized at 4 and 48 h for biodistribution. Radiolabeling was performed with a slight modification to published protocols [27]. Briefly, 0.5–1.0 mL of 5 mg/mL αCTLA-4 was buffer exchanged with 0.1 M sodium bicarbonate using Zeba™ Spin Desalting Columns, 7K MWCO, 5 mL (ThermoFisher). p-SCN-Bn-Deferoxamine (Macrocyclics, Plano TX, USA) was dissolved in DMSO to make a 3.8 mg/mL, and a 5 molar excess amount of the chelator was added to the buffer exchanged antibody. The solution was incubated at 37 °C for 1 h with gentle mixing, and then the buffer exchanged with a solution of freshly prepared 5 mg/mL gentisic acid with 0.25 M sodium acetate.

Aliquots of 1.5–1.7 mCi of ^89^Zr (Washington University at St. Louis, MO, USA) were diluted with 200 µL of freshly prepared 1 M oxalic acid, and 90 µL of 2 M sodium carbonate was gently mixed for 3 min at RT. The reaction was stopped by adding 300 µL of 0.5 M HEPES, followed by 200 µL of chelated antibody, 510 µL gentisic acid solution, and finally, an additional 700 µL of 0.5 M HEPES to yield a solution of 1 mg chelated antibody in 2 mL reaction mixture. The reaction mixture was incubated at RT for 1 h with gentle mixing. A radiolabeling efficiency of 99–100% was measured with ITLC (Biodex 150–771) and a running buffer of 20 mM citric acid. The chelated antibody was buffer exchanged with gentisic acid solution and aliquoted for injection. The presence of chelated antibody was verified using HPLC (Hitachi, Tokyo, Japan). At euthanasia, tissues were collected and radioactivity counted (2480 Wizard^2^ Automatic Gamma Counter (Perkin Elmer, Waltham, MA, USA). Counts were time-corrected and normalized to the activity of 50 μL of solution containing 0.025 mg of αCTLA-4 in order to report biodistribution in terms of % injected dose (ID)/mg of tissue of αCTLA-4. Since the excision of LNs incurs varying amounts of fat, tissues were uniformly dissected by one person (FCV) and the biodistribution was expressed as ng of αCTLA-4.

### 2.6. Flow Cytometry

Peripheral blood samples (500 µL) were treated with RBC lysis buffer (BD Biosciences, San Jose, CA, USA). Tumor tissues were gently ground and filtered through a 70 μm nylon mesh strainer (VWR, Radnor, PA, USA, Cat: 76327-100) to obtain a single-cell suspension. The cells were washed and resuspended in PBS, then stained with the following antibodies according to standard protocols. For surface staining, the cells were stained with the following markers at 4 °C: CD45, Pacific Blue anti-mouse CD45 (Biolegend, San Diego, CA, USA, Cat: 103126); CD3, Alexa Fluor 700 anti-mouse CD3 (Biolegend, Cat: 100216); CD4, BV650 anti-mouse CD4 (Biolegend, Cat: 100469); CD8a, Alexa Fluor 488 anti-mouse CD8a (Biolegend, Cat: 100723). For intracellular cytokine staining, the cells were stimulated with 10 µg/mL Melan-A pooled peptides or without (as unstimulated control) in the presence of 2 μM monensin (ThermoFisher, Cat: 00-4505-51) and 3 μg/mL Brefeldin A (ThermoFisher, Cat: 00-4506-51) at 37 °C for 16 h. For positive stimulation control, the cells were stimulated with 50 ng/mL phorbol myristate acetate (Millipore Sigma, Cat: P1585-1MG) and 1 μg/mL ionomycin (Millipore Sigma, Burlington, MA, USA Cat: 10634-1MG) in the presence of 2 μM monensin and 3 μg/mL Brefeldin A. After staining for surface markers, the cells were fixed, permeabilized, and stained with appropriate cytokine antibodies: TNFα, PECy7 anti-mouse TNFα (Biolegend, Cat: 506324), and IFNγ, Alexa Fluor 647 anti-mouse IFNγ (Biolegend, Cat: 505814) as per the manufacturer’s instructions. Isotype controls were used to subtract the background staining. Compensation adjustments were performed using BD^TM^ CompBead Plus (Cat # 560499). The stained cells were assessed using a BD FACS Aria II flow cytometer (BD Biosciences, San Jose, CA, USA), and data were analyzed with FlowJo V10 software (TreeStar Inc., Ashland, OR, USA).

### 2.7. Melan-A Specific Functionality

To determine the Melan-A specific T-cells, pooled 15-mer peptides of Melan-A (GenScript, Piscataway, NJ, USA) synthesized with 12 amino acids overlapping between neighbor peptides and covering the full length of the Melan-A protein were used during in vitro immune cell-stimulation (Stim). Non-stimulated (Non-Stim) controls were used to allow the determination of Melan-A specific T-cell (CD4^+^ or CD8^+^) functionality by measuring the percentage of cytokines (TNFα and IFNγ) in Non-Stim and subtracting from Stim: %CD4^+^TNFα^+^ (Stim)-%CD4^+^TNFα^+^ (Non-Stim).

### 2.8. Multiplex Immunofluorescence

For the immunoprofiling of tissues not used for flow cytometry, multiplex immunofluorescence (mIF) staining was performed using similar methods that have been previously described and optimized [28]. Briefly, four micrometer-thick formalin fixed, paraffin embedded sample sections were stained using a mouse mIF panel contained antibodies against: cytokeratin 19 (clone TROMA-III, Developmental Studies Hybridoma Bank, Iowa, IA, USA, Cat: AB-2133570), CD3 (clone D4V8L, Cell Signaling Technology, Danvers, MA, USA, Cat: 99940), CD8 (clone D4W2Z, Cell Signaling Technology, Danvers, MA, USA, Cat: 98941), CD4 (clone D7D2Z, Cell Signaling Technology, Cat: 25229), PD-1 (clone D7D5W, Cell Signaling Technology, Cat: 64988), PD-L1 (clone D5V3B, Cell Signaling Technology, Cat: 64988), and F4/80 (clone D2S9R, Cell Signaling Technology, Cat: 70076). All the markers were stained in sequence using their respective fluorophore contained in the Opal 7 IHC kit color (catalogue #NEL797001KT; Akoya Biosciences, Waltham, MA, USA; Cat: NEL797001KT) and the individual tyramide signal amplification fluorophore Opal Polaris 480 (Akoya Biosciences, Marlborough, MA, USA, Cat: FP1500001KT). The slides were scanned using the Vectra/Polaris 3.0.3 (Akoya Biosciences) at low magnification, 10× (1.0 µm/pixel) through the full emission spectrum and using positive mouse lymph node controls from the run staining to calibrate the spectral image scanner protocol. A pathologist (LCC) selected at least five regions of interest (ROIs) for scanning in high magnification using the Phenochart Software image viewer 1.0.12 (931 × 698 µm size at resolution 20×) in order to capture various elements of tissue heterogeneity. Each ROI was analyzed by a pathologist using InForm 2.4.8 image analysis software (Akoya Biosciences). The densities of each cell phenotype were quantified, and the final data were expressed as the number of cells/mm^2^. All the data were consolidated using the R studio 3.5.3 (Phenopter 0.2.2 packet, Akoya Biosciences). 

### 2.9. Statistics

For the orthotopic tumor model, data were analyzed using unpaired, two-tailed Student’s *t* test unless otherwise noted. For the metastatic tumor model, the data were analyzed using a one-way analysis of variance (ANOVA) model with Fisher’s least significant difference method for pair wise comparisons. The survival analysis was conducted using the Kaplan-Meier curve and the log rank test.

## 3. Results

### 3.1. CBI Delivery to the Right Lymphatic Watershed following i.d. Injection at Right Lateral Side of Tail

I.d. administration of CBI to the right base of the tail consistently transited unilaterally to the right popliteal and inguinal LNs and, eventually, to the right axillary LNs as shown from the NIRF imaging (Figure 3A and Appendix A). Biodistribution results show that significantly greater drug initially reached the right inguinal and axillary LNs at 4 h after i.d. administration when compared to i.v. administration. At 48 h, comparable amounts were found in the LNs of animals dosed i.v. and i.d., although there was consistently less drug in the non-tdLNs (left side) as opposed to the tdLNs (right side) after i.v. administration (Figure 3B). The latter may be due to increased drainage of drug accumulated at the tumor site. Lymphatic delivery resulted in reduced blood and organ drug levels at 4 h when compared to i.v. delivery; although by 48 h, there was no difference, consistent with the unidirectional delivery of drug from the lymphatics to the hemovascular system (Figure 3C).

### 3.2. Tumor Growth Rates Significantly Reduced with i.d. but Not i.v. Delivery of CBI Monotherapy in Vaccinated Animals

As shown in Figure 4A,B, non-vaccinated animals receiving systemic, i.v. injection of αCTLA-4 or αPD-1 monotherapy experienced no change in tumor growth rates compared to the control (i.e., no treatment), consistent with prior reports in the literature [29,30,31,32]. However, reductions in average tumor sizes occurred in animals receiving αCTLA-4 monotherapy, i.d. (*p* < 0.05) starting at 15 d.p.i., and in animals receiving αPD-1 monotherapy i.d., starting at 14 d.p.i. Vaccination with Melan-A VLP alone (Figure 4C) or in combination with CBI significantly reduced the average tumor size. It is noteworthy that VLP vaccination increased the number of animals responding to treatment with i.d. αCTLA-4 administration (Appendix A). Interestingly, when animals were vaccinated with non-specific Gag-VLP, the therapeutic advantage of lymphatic dosing of αCTLA-4 was lost, as tumor sizes were similar to the control group and greater than the treatment groups (isotype IgG2b, i.d.; isotype IgG2b, i.d. + Melan-A VLP; αCTLA-4, i.d. + Melan-A VLP; and αCTLA-4 i.d.; Figure 4C and Appendix A).

Together, these data provide compelling evidence that lymphatic, as opposed to systemic, administration of CBI monotherapy results in more effective anti-tumor responses both with and without Melan-A VLP vaccination.

### 3.3. Melan-A VLP Vaccination and Lymphatic Delivery of CBI Impacts the TILs

Flow cytometry showed significant increases in %CD45^+^CD3^+^ tumor-infiltrating lymphocytes (TILs) in unvaccinated animals receiving i.d. administration of αCTLA-4 as compared to animals receiving no treatment (Figure 5A), yet there were no differences in the %CD45^+^CD3^+^ PBMCs (Appendix A). Few TILs were found in unvaccinated animals treated with αPD-1 i.d. or i.v. administration (Figure 5B) although there were greater %CD45^+^CD3^+^ cells in the PBMCs of animals dosed with αPD-1 i.d. than i.v. (Appendix A). The addition of Melan-A VLPs significantly increased the TILs found from flow cytometry in animals receiving i.d. αCTLA-4 and αPD-1 monotherapies (but not systemically), even though there were no changes in %CD45^+^CD3^+^ PBMCs. MIF analysis confirmed increases in the TIL density in vaccinated animals dosed with αCTLA-4 compared to the control animals (see Appendix A). Melan-A VLP alone did not significantly increase TILs (Figure 5C) or lymphocytes in PBMCs (Appendix A); combination with αCTLA-4 (i.d. or i.v.) or αPD-1 (i.d.) was necessary to significantly increase the TILs (Figure 5A,B). 

Compared to the controls, the population of CD4^+^ PBMCs increased in vaccinated animals, while CD4^+^ TILs in unvaccinated animals receiving i.d. and i.v. αCTLA-4 tended to decrease (Figure 6A, Appendix A), consistent with reports of antibody-dependent cellular cytotoxicity (ADCC) action against αCTLA-4 expressing CD4^+^ Tregs in the TME [33]. In contrast, CD4^+^ TIL populations found from flow cytometry and mIF increased in vaccinated animals receiving αCTLA-4 i.d. and i.v., consistent with VLP induced MHC class II presentation of Ag that was promoted by CD4^+^ T-cell priming through blockage of CTLA-4 signaling. Melan-A VLP vaccination and boost alone reduced CD4^+^ TILs without changing CD4^+^ PBMCs (Figure 6B, and Appendix A), possibly due to a reduction in Treg population in the TME [34] and a lack of sufficient priming and maintenance of CD4^+^ cell priming against TAA. Although vaccination alone tended to increase CD8^+^ PMBCs and TILs compared to the control groups, when combined with αCTLA-4, a significant reduction in both CD8^+^ and TIL populations were found from flow cytometry (Figure 6B, Appendix A). A trend of reduced CD8^+^ and increased CD4^+^ TILs was also noted from mIF (Appendix A). This result may be explained by systemic tolerance against activated CD8^+^ primed against TAA through PD-1/PDL-1 and other co-inhibitory molecules. In contrast, vaccinated animals receiving αPD-1 lymphatically exhibited significantly greater CD8^+^ TILs and PBMCs (Figure 6C, Appendix A), presumably due to efficient VLP-induced MHC I cross presentation of TAA and the added arrest of tolerization from PD-1/PD-L1 signaling in the lymphatics.

The lack of TILs limited the characterization of how lymphatic versus systemic delivery of monotherapy may have impacted TAA specificity and the TME. The TAA functionality of PBMCs in vaccinated animals at the time of euthanasia was unremarkable. In summary, the results point to αCTLA-4 monotherapy increasing CD4^+^ TILs in vaccinated animals by enhancing VLP induced MHC II cross presentation of TAA in the LN, while αPD-1 monotherapy increases CD8^+^ TILs by interrupting PD-1/PD-L1 signaling that would otherwise tolerize MHC I cross presentation of TAA.

### 3.4. Tumor Irrelevant Vaccination Creates a Pro-Tumor TME

When tumor irrelevant vaccination against the SIV gag was combined with αCTLA-4 i.d., we found a significant increase in TILs from flow cytometry compared to the control and treated animals (no treatment, IgGb2 i.d., and IgGb2, i.d. + Melan-A VLP) (Figure 5C) without any change in PBMC lymphocytes (Appendix A). Increases in CD4^+^ and decreases in CD8^+^ populations were observed in the TME of animals treated with Gag-VLP and αCTLA-4, i.d. (Figure 6B). However, vaccination against gag did not increase CD4^+^ PBMCs as it did in the TME (Appendix A). As shown in the mIF analyses, elevated CD4^+^ populations were found in the TME of irrelevantly vaccinated animals dosed with αCTLA-4, i.d. (Appendix A).

Taken together with the loss of therapeutic benefit from i.d. dosing of αCTLA-4 in animals vaccinated with Gag-VLP (shown above in Figure 4C), our results evidence the importance of tAg or TAA specific T-cell priming. Most importantly, these results point to the detrimental aspects of Ag-indiscriminate T-cell activation that could potentially result from systemic administration of CBIs. This non-specific T-cell activation could account for the low efficacy and significant toxicity often experienced by cancer patients who received αCTLA-4 therapy.

### 3.5. Tumor Growth Rates Significantly Reduced with i.d. Delivery of αCTLA-4 in Combination Therapy

While the blockage of CTLA-4 to enhance the initiating step of tumor-specific T-cell priming occurs in the tdLNs, two sites for αPD-1 action are (1) in the tdLN where PD-1 is upregulated with TCR signaling, and (2) in the TME where PD-L1 expression on tumor cells and macrophages can ligate PD-1 on otherwise cytotoxic CD8^+^ and CD4^+^ T-cells (Figure 1). Thus, to test the relative importance of lymphatic versus systemic αPD-1 administration, we compared animals dosed with αCTLA-4 i.d. combined with αPD-1 dosed i.v. or i.d. Compared to the control, both vaccinated and unvaccinated animals receiving αCTLA-4 i.d. and αPD-1 i.v., but not those receiving combination CBI i.d. or i.v., had significantly reduced tumor sizes at 17 d.p.i. (Figure 7A,B). Previous genetic ablation studies show that PD-1 loss compromises the proliferative hierarchy of cells early after TCR signaling and leads to increased apoptosis during the contraction phase and upregulation of other inhibitory receptors (Lag-3, Tigit, CD160, 2B4/CD244) that can temper T-cell responses [35]. In vaccinated animals receiving combination CBI wholly through the lymphatic route, at 17 d.p.i. we found trends of reduced TILs from flow cytometry and from mIF (Figure 7C and Appendix A) consistent with terminal differentiation associated with a lack of PD-1 at early TCR signaling potentiated by VLP vaccination. With the exception of vaccinated animals receiving combinational therapy lymphatically, flow cytometry showed TILs in vaccinated animals receiving combinational CBI were comparable to that of vaccinated animals receiving lymphatic dosing of monotherapies. Flow cytometry further showed increased CD4^+^ TILs in vaccinated animals receiving αCTLA-4 lymphatically or systemically (Figure 7D) over unvaccinated animals. This result is consistent with VLP induced MHC II cross presentation in the LN. The greatest percentage of CD8^+^ TILs occurred in unvaccinated animals receiving αCTLA-4 i.d. and αPD-1 i.v., while there were fewer CD8^+^ TILs found in vaccinated animals receiving combination CBI; again, possibly indicative of overstimulation and terminal CD8^+^ differentiation. Nonetheless, we observed significant increases in Melan-A specific CD4^+^ and CD8^+^ PBMC populations in vaccinated animals receiving αCTLA-4 i.d. and αPD-1 i.v., but not when combinational therapy was administered either entirely systemically or entirely lymphatically (Figure 8A). The IHC of tumors of vaccinated animals receiving combined CBI i.d. shows reduced TILs, CD4^+^ TILs, and macrophage infiltration, but enhanced PD-L1^+^ macrophages when compared to vaccinated animals receiving combined CBI i.v. (Figure 8B). As shown in Figure 8C, the density of cells expressing PD-L1^+^ cells (T-cells, CK19^+^ tumor cells, and predominantly macrophages) were elevated in animals receiving combination CBI wholly through i.v. and i.d. routes, but not when αCTLA-4 and αPD-1 were respectively administered i.d. and i.v. In addition, CK19 was validated as a melanocyte marker in B16F10 cells.

Interestingly, Kaplan-Meier survival statistics show that vaccinated animals receiving the combination of αCTLA-4/αPD-1 lymphatically had significantly increased survivorship, while the survival of vaccinated animals receiving both αCTLA-4 lymphatically and αPD-1 systemically did not reach significance (Figure 8D). There was no significant difference between the survivorship of vaccinated animals receiving combinational therapy with αPD-1 lymphatically or systemically. Whether the generation of memory T-cells upon terminal expansion is caused by lymphatic delivery of CBIs for potent immunological responses after VLP vaccine boost remains to be investigated. Nonetheless, the data on lymphatic delivery of CBIs result is consistent with (i) the role of αCTLA-4 in effective T-cell priming, (ii) the mechanism of action of αPD-1 inside the lymphatics, after TCR signaling, and within the TME, and potentially, (iii) the drainage of both CBIs administered lymphatically into the systemic circulation.

It is noteworthy that when αCTLA-4 was administered i.d. to the tdLN, and αPD-1 delivered i.v., there was little difference in survival between vaccinated and unvaccinated animals (*p* = 0.9998). Given that the Melan-A epitope on the VLP was developed from the B16F10 cell line, the delivery of αCTLA-4/αPD-1 to tdLNs may be sufficient to initiate and, in the short time frames investigated herein, sustain the immune response equally well in unvaccinated as in vaccinated animals. One may speculate that survival may be further enhanced using multi-targeted VLPs using neoantigens, as opposed to TAAs, to elicit longer-lived, tumor specific immune responses [22,36]. Nonetheless, we showed that when systemic combinational CBI therapy was combined with VLP, tumor growth rates were significantly reduced compared to untreated animals, but still higher than unvaccinated animals receiving αCTLA-4 i.d. and αPD-1 i.v., indicating the fundamental T-cell priming role of αCTLA-4 in the tdLN. Because systemic administration of CBIs may cause Ag indiscriminate T-cell activation outside of the lymphatic watershed receiving vaccination, bystander T-cells may contribute to a pro-tumor TME negating the impact of CBIs.

### 3.6. Lymphatic Delivery of CBIs Decreases Metastatic Potential in an Orthotopic Melanoma Mouse Model

To evaluate lymphatic versus systemic deliveries in a potentially more relevant metastatic model, we delivered CBIs lymphatically and systemically in vaccinated and unvaccinated animals bearing orthotopic tumors beginning at 7 d.p.i. Visible metastases were present in the popliteal LN at 7–9 d.p.i. in approximately 50% of the untreated animals, suggesting a pro-tumor tdLN environment had been established at the time of CBI initiation. A relative mRNA expression of 10^4^ corresponded to approximately 10^6^ B16F10 melanoma cells, suggesting it as a sensitive measure of metastatic tumor progression. Blood mRNA levels of *Pmel17* increased from baseline to seven orders of magnitude in untreated animals (Figure 9A) with the highest signals in the inguinal tdLN, followed by blood, the axillary tdLN, and lastly the contralateral inguinal LN, as determined at 21 d.p.i. (Figure 9B). The results are consistent with the dissemination of cancer through the lymphatics into the hemovascular system. Figure 9C i–iii illustrate the change in relative *Pmel17* expression as a function of time showing improved anti-tumor responses in Melan-A VLP vaccinated animals with maximal anti-tumor responses in vaccinated animals receiving lymphatic delivery of combination therapy by 18 d.p.i. In contrast to subcutaneous (s.c.) implants, there was little to no primary tumor implants to evaluate the TILs. Nonetheless, the results in orthotopic tumors provide another validation that lymphatic delivery significantly improves the efficacy of CBIs.

## 4. Discussion

While CBI can provide durable anti-tumor immunity, the majority of cancer patients experience low responses and/or exhibit significant toxicities that limit its use. Clinical trials to combine CBI with regional radiation, therapeutic vaccinations, and systemic chemotherapies promise to improve efficacy through altering the TME by increasing tAg, antigenic CD8^+^ and CD4^+^ Th1 effector cell populations and minimizing bystander T-cells or other infiltrating cells that promote a pro-tumorigenic environment. However, these therapeutic combinations may not adequately address the underlying indiscriminate Ag specificity associated with systemic administration of CBIs. Even though adaptive immune responses are well-known to be established within dLNs, CBIs are administered to achieve systemic therapeutic dosage levels, potentially contributing to Ag-indiscriminate T-cell activation that can lead to pro-tumorigenic bystander T-cells. The European Society for Medical Oncology (ESMO) position statement on CBI dosing focuses upon reducing Ag-irrelevant T-cell responses through local administration by dosing CBIs peri- or intra-tumorally. Their idea is that tAg cross presentation to tumor resident naïve T-cells and ADCC of immunosuppressive cell populations within the TME can be better targeted without inducing Ag-irrelevant responses that most likely occurs with systemic administration [37]. However, adaptive immune responses occur within the close confines of dLNs, where activated CD4^+^ T-cells secrete cytokines that drive CD8^+^ effector function, differentiation, and proliferation; induce CD8^+^/APC interaction; and direct humoral responses. The expanse of the TME may not contain the concentrated assemblage of immune players and mediators that are otherwise secluded and protected within the LN environment to efficiently orchestrate immune responses. In addition, tumor-draining lymphatics are often dysfunctional [38]. As a result, systemic, peri- or intra-tumoral administration may not result in maximal exposure to targets within the lymphatics and dLNs necessary for T-cell priming. In addition, while most immunotherapies focus upon cytotoxic CD8^+^ populations, the role of CD4^+^ Th1 cytotoxic and helper functions is increasingly recognized as necessary to drive anti-tumor immunity [39,40]. Because the full assembly of immune components for CD4^+^ and CD8^+^ activation are found within LNs, our results demonstrating how lymphatic delivery of CBIs improve anti-tumor responses are consistent with classical teachings on the role of LNs in adaptive immunity.

Upon using radiolabeled αCTLA-4, or by spiking CBI doses with the ICG fluorescent contrast used by us in clinical studies [38], we showed that i.d. administration of CBIs at the base of the tail reaches the ipsilateral popliteal, inguinal, and axillary tdLNs that also drain the B16F10 melanoma implant. Although the i.d. and i.v. routes resulted in similar biodistribution of drug at late times after administration, i.d. administration resulted in a greater anti-tumor response with and without vaccination. This may be due to immunodominance or the “priority of first response”, whereby Ags inducing a faster response become immunodominant, outcompeting the subsequent cross presentation of other sub-immunodominant Ags [41].

We purposely used the B16F10 melanoma cell line that is well-known to result in a non-immunogenic tumor model and is un-responsive to systemic administration of αCTLA-4 and αPD-1 monotherapies. We demonstrated that the average tumor size, as a read-out of anti-tumor immunity, was reduced with lymphatic delivery of the CBI monotherapy to tdLNs in unvaccinated animals. The tdLNs were likely natural recipients of tumor-derived Ags. Despite the reduction in average tumor growth rates with αCTLA-4, i.d. monotherapy, a group of non-responders persisted (Appendix A), which may be due to tumor immune escape, or a loss of MHC in the TME, or an already tolerized tdLN that restricted the successful and prolonged activation of T-cells against tAgs. Because no discernable difference in TILs or PBMCs occurred between lymphatic and systemic administration, the enhancement of responses from lymphatic delivery in unvaccinated animals is likely due to the improved Ag specificity of TILs. Future work will assess the TCR repertoire and diversity of TILs to underscore the role of lymphatic delivery to dLNs in shaping the TME. It is also noteworthy that anti-tumor responses were further improved when lymphatic delivery of αCTLA-4 was combined with lymphatic or systemic administration of αPD-1 in unvaccinated animals, further evidencing the importance of tAg-specific T-cell priming in the tdLNs and the role of co-inhibitory signaling in the lymphatics and the TME. However, we cannot rule out the possibility that dosing combination CBIs with VLP in tdLNs could result in overstimulation and terminal differentiation of cytotoxic CD8^+^ T-cells as described in PD-1 ablation studies of viral infection [35]. Nonetheless, the survival analysis of the s.c. model at 30 d.p.i. (Figure 8D) and the response of the orthotopic model (Figure 9C) suggest lymphatic delivery of combination CBI positively influenced the immune responses, consistent with the enhancement of memory T-cells following terminal differentiation [35].

VLP vaccines provide opportunities for MHC I and II cross presentation of tAg to naïve CD4^+^ and CD8^+^ T cells [22], with i.d. administration providing maximal cellular and humoral immunity compared to i.v. or s.c. administration [42]. I.d. administration enables direct entry of VLPs into dLNs as well as uptake by epidermis-resident Langerhans cells that, once activated, mature into APCs as they transit through afferent lymphatic vessels for tAg cross presentation within dLNs [42]. Clinically, VLP-based vaccines against TAA Melan-A have been tested in melanoma patients showing Melan-A/Mart-A specific CD8^+^ and CD4^+^ responses in the majority of patients with sustained immune responses lasting more than a year [22]. Using “click-chemistry” to couple multiple peptides to TLR 9 ligand-loaded VLPs, Bachmann and coworkers showed VLPs can be designed as personalized cancer vaccines for targeting mutated and germline epitopes specific for melanoma [36]. In this preclinical work, we show that Melan-A VLP increased CD4^+^ TIL populations in animals dosed with αCTLA-4; increased CD8^+^ TILs in animals dosed lymphatically with αPD-1; and decreased CD8^+^ and increased CD4^+^ TILs when combinational treatment was dosed lymphatically. When VLP delivered tumor-irrelevant Ag, TILs increased with lymphatic delivery of αCTLA-4, but the therapeutic advantage of αCTLA-4 was lost. This data further evidences the importance of the antigen specificity of T-cell activation for anti-tumor immunity and preventing immune toxicity. Vaccinated animals receiving lymphatic combinational treatment exhibited significantly reduced tumor sizes, improved survival, greater Melan-A functionality of CD8^+^ and CD4^+^ PBMCs, and, in an orthotopic model, decreased the metastatic burden as indicated by blood *Pmel17* RNA expression, than when dosed systemically. Given that LECs upregulate MHC machinery for Ag presentation along co-inhibitory molecules [43], the lymphatic delivery of αPD-L1/3, αLAG-3 may additionally target LEC-driven inhibition in afferent lymphatic vessels tdLNs that are not reached with i.v. administration. Because lymph empties into the blood circulation, lymphatic delivery of αPD-L1 may additionally reach targets outside the lymphatics, namely those in the TME to reverse signaling that tolerizes T-cells.

It is important to note that the Melan-A/Mart antigen used herein is a TAA. As a result, Melan-A-specific T-cells can be subject to elimination by central tolerance mechanisms, thereby limiting efficacy and resulting in heterogenous responses. In addition, allelic MHC I loss by tumor cells under immune pressure can result in futile T-cell priming and additional mechanisms of tumor immune escape [41]. Multi-target vaccination and the selection of multiple tumor neoantigens could limit tumor immune escape [22,36].

There are several limitations and caveats to this preclinical study. First, this study was performed in only two B16F10 melanoma models to compare results of s.c. models found in the literature [29,30,31,32], and to evaluate changes in metastatic tumor burden in a less common tumor model. Our prior work in immunogenic 4T1-luc models is consistent with B16F10, showing improved efficacy of αCTLA-4 when delivered lymphatically over systemic administration [44]. FACS evaluation of other immune cells and Melan-A functionality of T-cells in the TME was limited by cell numbers. Future work to evaluate how lymphatic delivery and VLP vaccination change the TCR repertoire and transcriptomic phenotypes of TILs using techniques such as single cell RNA sequencing may better demonstrate the mechanisms behind how pharmacological manipulation of dLNs can impact the immune status of the TME. Finally, TME assessment by flow and mIF was biased against complete responders and were conducted from separate animal cohorts. Nonetheless, this work shows that delivery of tAg and CBI to the lymphatics impacts the tumor immune microenvironment.

It is important to point out that the lymphatic volume of rodents is considerably smaller than in humans. As such, it is likely that the lymphatic dosing of CBIs resulted in exposure to targets in the lymphatics and outside the lymphatics, including the TME. Lymphatic dosing in bipedals (humans, non-human primates) may be more compartmentalized to a lymphatic watershed than rodent quadrapedals. If tAg-specific T-cell priming and maintenance is proven to be the initiating, primary driver of therapeutic efficacy, then CBI doses could be reduced to maximize exposure to dLNs and minimize systemic exposure, thereby reducing tAg-indiscriminate immune responses and potentially reducing severe immune-related adverse toxicities. Further study is needed to determine whether regional delivery to tdLNs can reduce irAEs by limiting systemic exposure. Because wild-type mice do not provide reliable readouts that can indicate clinical irAEs, future studies are needed in transgenic models of transient Treg depletion [45] to show reduction in antigen-indiscriminate T-cell expansion and infiltrating lymphocytes in normal tissues with regional, lymphatic as opposed to systemic delivery of CBI. Because Treg depletion will likely enhance responses to CBI, these studies of irAEs are best performed in non-tumor bearing animals. Clinically, tdLNs are frequently the subject of pre-treatment (i.e., surgical dissection or radiation). By combining lymphatic delivery with tAg-VLP vaccination in non-tdLNs that are not already cancer tolerized or impaired by cancer treatment, it is possible that equivalent, if not better, anti-tumor CBI responses will result. Finally, it is important to note that i.d. administration volumes through conventional hypodermic needles are clinically limited to 100 µL or less, potentially limiting the direct delivery of CBI to dLNs. Future development of microneedle arrays for accurate i.d. infusion of clinically relevant amounts and/or volumes of CBI is underway by several academic laboratories and drug delivery companies [46,47]. Alternatively, s.c. administration or implantation of s.c. drug-eluting devices within an intact lymphatic watershed could also reach LNs, although with less bioavailability than via i.d. administration.

## Figures and Tables

**Figure 1 cancers-14-01823-f001:**
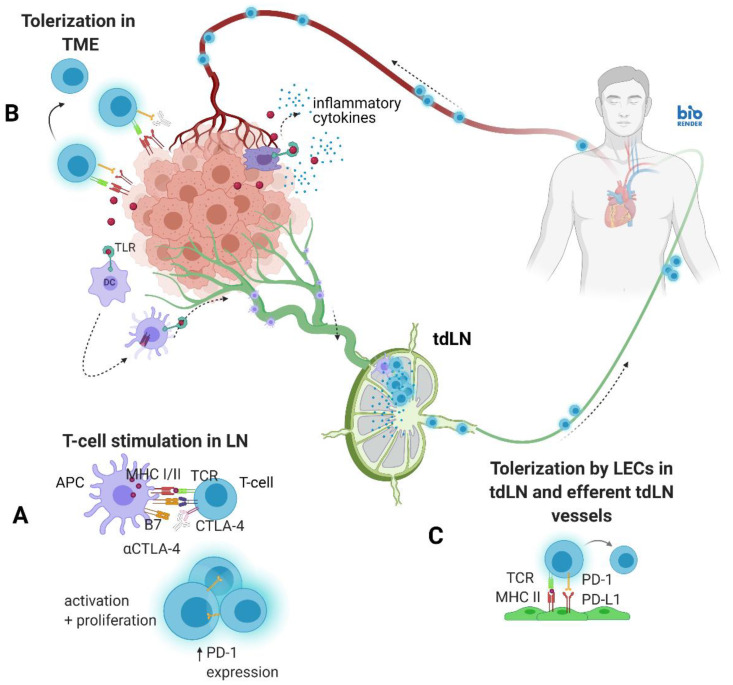
Lymph flow from tumor draining lymph nodes (tdLNs) to blood circulation. (**A**) Presentation of tumor antigen (tAg) by antigen presenting cells (APCs) to naïve T-cells in tdLNs initiates anti-tumor response. (**B**) Activated T-cells can be tolerized by PD-L1 expression within an immunosuppressive tumor microenvironment (TME). (**C**) Lymphatic endothelial cells (LECs) may also present tAg and express high levels of PD-L1 to tolerize effector T-cells in tdLNs and efferent lymphatic vessels. TCR, T cell receptor; MHC, major histocompatibility complex; LN, lymph node. Created with Biorender.com (accessed on 21 February 2022).

**Figure 2 cancers-14-01823-f002:**
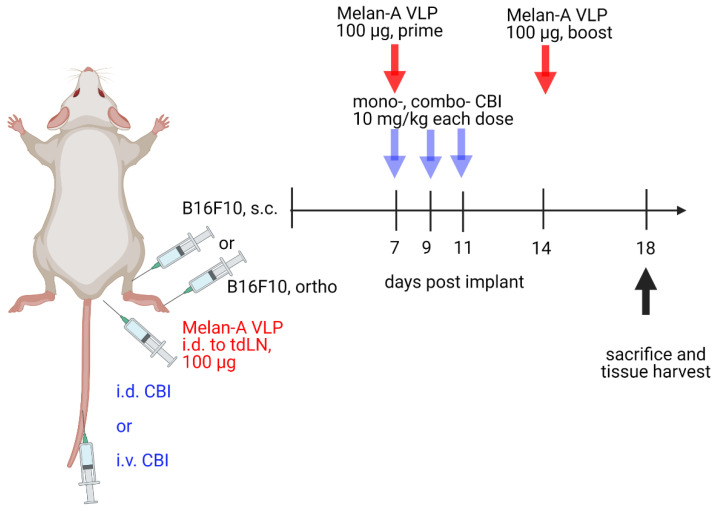
Orthotopic mouse models for melanoma B16F10 cell inoculation in hindlimb and dorsal aspect of hind paw and timeline for checkpoint blockade immunotherapy (CBI) dosing and sacrifice. i.d., intradermal; i.v. intravenous; s.c., subcutaneous; ortho, orthotopic. Created with Biorender.com (accessed on 21 February 2022).

**Figure 3 cancers-14-01823-f003:**
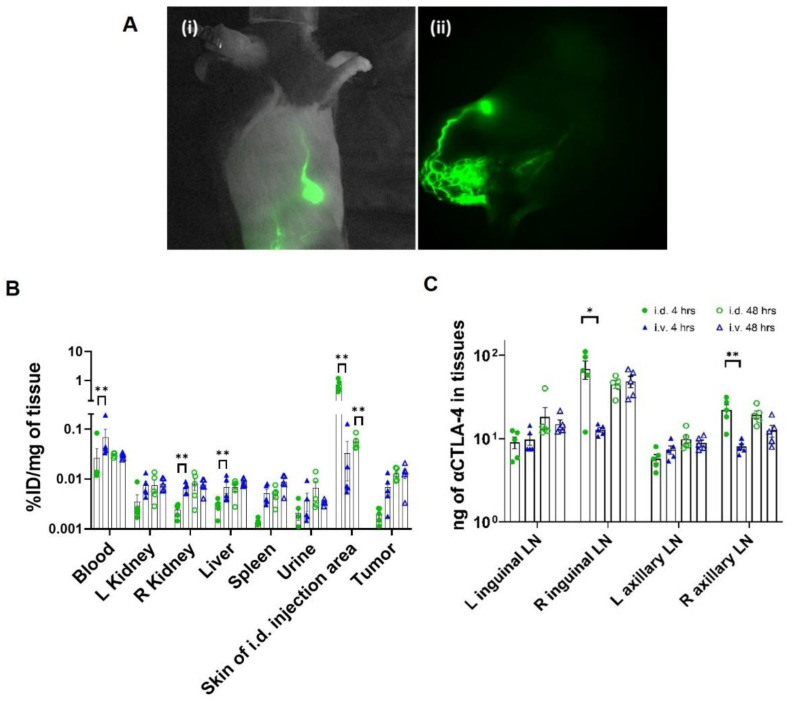
(**A**) Typical visualization of lymphatic drug delivery to (**i**) inguinal lymph node (LN) and subsequent delivery to axillary LN in mouse at 9 days post implant (d.p.i.) in ipsilateral lateral view (see Appendix A transit immediately after i.d. administration), and (**ii**) lymphatics draining popliteal and inguinal LNs at 21 d.p.i. in ventral view (see Appendix A). Appendix A show lymphatic propagation. Distribution of ^89^Zr-labeled αCTLA-4 in (**B**) LNs and (**C**) organs following i.d. dosing at 4 h (●, green, *n* = 5) and at 48 h (○, green, *n* = 5), i.v. dosing at 4 h (▲, blue, *n* = 5) and at 48 h (△, blue *n* = 5); *, ** Denotes statistical significance via unpaired, two-tailed student *t*-test (*p* < 0.05, 0.005). Error bars indicate SEM. ID, injected dose; L, left; R, right; i.d., intradermal; i.v., intravenous.

**Figure 4 cancers-14-01823-f004:**
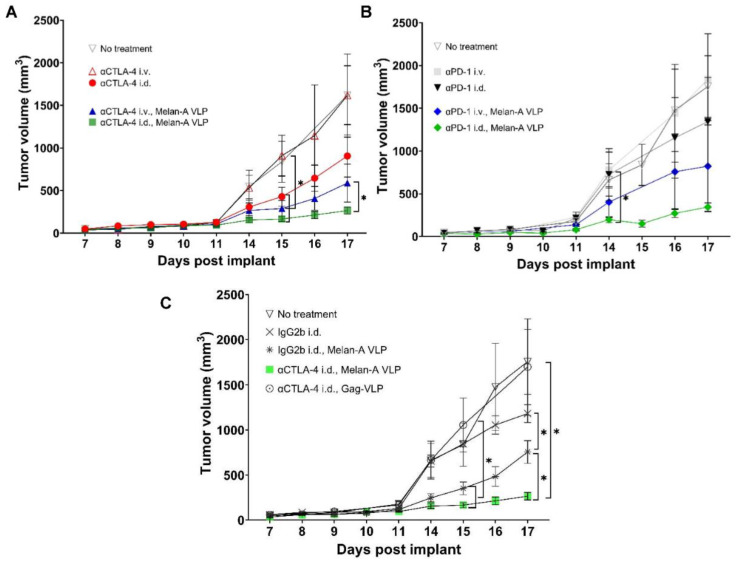
(**A**) Lymphatic delivery of αCTLA-4 monotherapy alone or in combination with Melan-A VLP significantly reduces tumor growth. Primary tumor volume versus d.p.i. for untreated animals (▽, grey *n* = 5); and animals dosed with αCTLA-4 i.v. (△, red, *n* = 5); αCTLA-4 i.d. (●, red, *n* = 7); αCTLA-4 i.v., Melan-A VLP (▲, blue, *n* = 10); and αCTLA-4 i.d., Melan-A VLP (■, green *n* = 10). (**B**) Lymphatic delivery of αPD-1 monotherapy alone or in combination with Melan-A VLP significantly reduces tumor growth. Primary tumor volume versus d.p.i. for untreated animals (▽, grey *n* = 5); and animals dosed with αPD-1 i.v. (■, grey *n* = 4); αPD-1 i.d. (▼, black, *n* = 9); αPD-1 i.v., Melan-A VLP (♦, blue, *n* = 4); and αPD-1 i.d., Melan-A VLP (♦, green, *n* = 10). (**C**) Lymphatic delivery of Melan-A VLP alone significantly reduces tumor growth. Primary tumor volume versus days post implant for untreated animals (▽, grey *n* = 5); and animals dosed with IgG2b i.d. (X, grey *n* = 4); IgG2b i.d., Melan-A VLP (✱, grey, *n* = 4); αCTLA-4 i.d., Melan-A VLP (■, green *n* = 10); and αCTLA-4 i.d., Gag-VLP (○, grey *n* = 10). * Denotes first time point of statistical significance from unpaired, two-tailed Student’s *t* test (*p* < 0.05). Error bars indicate SEM. i.v., intravenous; i.d., intradermal.

**Figure 5 cancers-14-01823-f005:**
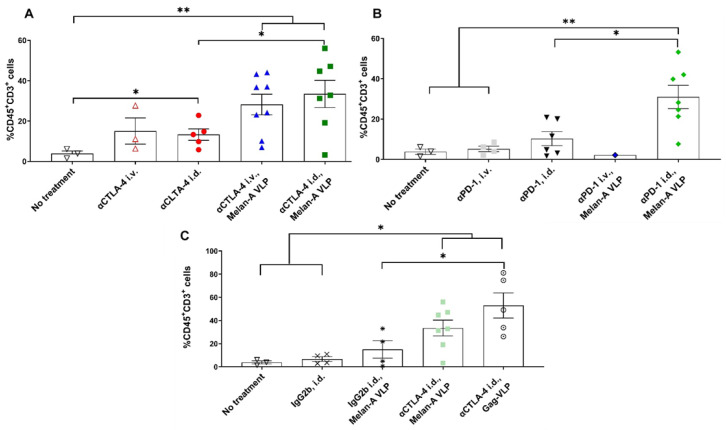
The % CD45^+^CD3^+^ in TME (TILs). (**A**) Lymphatic delivery of αCTLA-4 monotherapy and in combination with Melan-A VLP increases %CD45^+^CD3^+^ T-cells in the TME. No treatment (▽, black *n* = 3); αCTLA-4 i.v. (△, red, *n* = 3); αCTLA-4 i.d. (●, red, *n* = 5); αCTLA-4 i.v., Melan-A VLP (▲, blue, *n* = 8); and αCTLA-4 i.d., Melan-A VLP (■, green *n* = 7). (**B**) Lymphatic delivery of αPD-1 in combination with Melan-A VLP increases CD45^+^CD3^+^ T-cells in the tumor microenvironment. Few TILs were found in animals dosed with VLP and αPD-1 i.v. No treatment (▽, black *n* = 3); αPD-1 i.v. (■, grey *n* = 4), αPD-1 i.d. (▼, black *n* = 6); αPD-1 i.v., Melan-A VLP (♦, blue, *n* = 1); αPD-1 i.d., Melan-A VLP (♦, green, *n* = 7). (**C**) Melan-A VLP and Gag-VLP vaccination with lymphatic delivery of αCTLA-4 increases CD45^+^CD3^+^ T-cells in the tumor microenvironment. No treatment (▽, black, *n* = 3); IgG2b i.d. (X, black, *n* = 4); IgG2b i.d., Melan-A VLP (✱, *n* = 4); αCTLA-4 i.d., Melan-A VLP (■, green *n* = 7); αCTLA-4 i.d., Gag-VLP (○, black *n* = 5). *, ** Denotes statistical significance using unpaired, two-tailed Student’s *t* test (*p* < 0.05, <0.005). Error bars indicate SEM. i.d., intradermal; i.v. intravenous.

**Figure 6 cancers-14-01823-f006:**
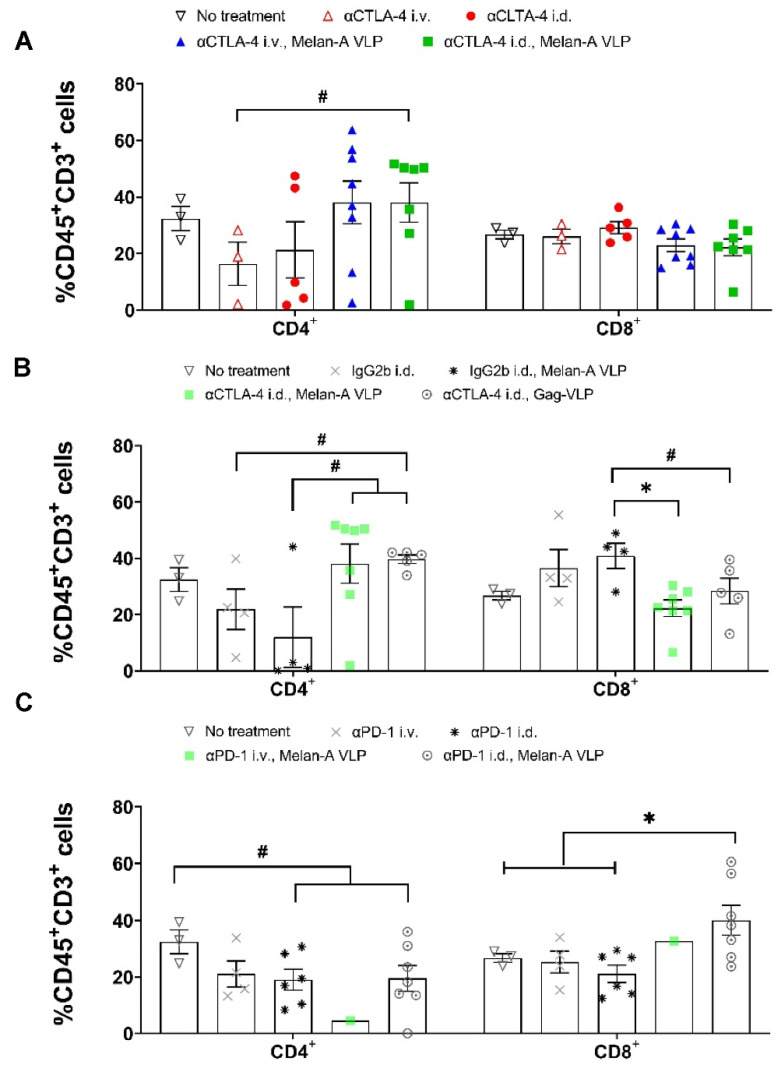
The %CD4^+^ and CD8^+^ cells in TME. (**A**) Delivery of αCTLA-4 monotherapy and in combination with Melan-A VLP changes the distribution of CD4^+^ T-cells in the tumor microenvironment. No treatment (▽, black, *n* = 3); αCTLA-4 i.v. (△, red, *n* = 3); αCTLA-4 i.d. (●, red, *n* = 5); αCTLA-4 i.v., Melan-A VLP (▲, blue, *n* = 8); and αCTLA-4 i.d., Melan-A VLP (■, green, *n* = 7). (**B**) Lymphatic delivery of αPD-1 in combination with Melan-A VLP changes the distribution of CD4^+^ and CD8^+^ T-cells in the tumor microenvironment. No treatment (▽, black, *n* = 3); αPD-1 i.v. (■, grey, *n* = 4); αPD-1 i.d. (▼, black, *n* = 6); αPD-1 i.v., Melan-A VLP (♦, blue, *n* = 1); and αPD-1 i.d., Melan-A VLP (♦, green, *n* = 7). (**C**) Effect of Melan-A VLP and Gag-VLP in the percentage of CD4^+^ and CD8^+^ T-cells in the tumor microenvironment. No treatment (▽, black, *n* = 3); IgG2b i.d. (X, grey, *n* = 4), IgG2b i.d., Melan-A VLP (✱, *n* = 4); αCTLA-4 i.d., Melan-A VLP (■, green, *n* = 7); and αCTLA-4 i.d., Gag-VLP (○, *n* = 5). Statistical significance by unpaired, two-tailed Student’s *t* test (*p* < 0.05) is denoted by * and unpaired, one-tailed Student’s *t* test (*p* < 0.05) is denoted by #. Error bars indicate SEM. i.d., intradermal; i.v. intravenous.

**Figure 7 cancers-14-01823-f007:**
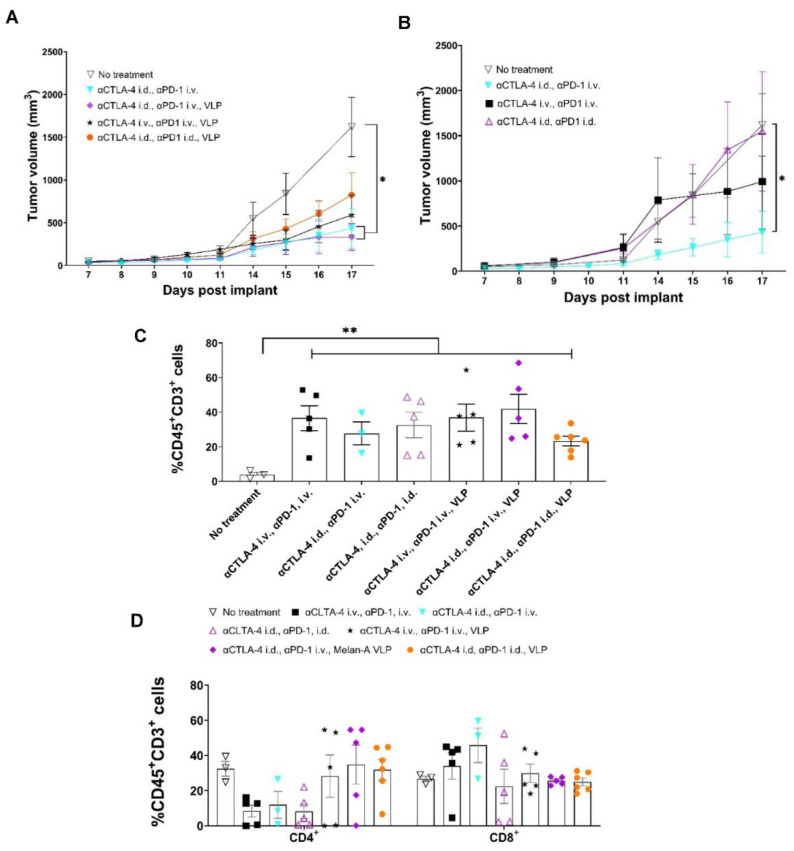
Response to lymphatic and systemic combination treatment in vaccinated animals. (**A**) Lymphatic delivery of αCTLA-4 and Melan-A VLP in combination with αPD-1 further decreases primary tumor growth. No treatment (▽, *n* = 5); αCTLA-4 i.d., αPD-1 i.v., (▼, blue, *n* = 5); αCTLA-4 i.v., αPD-1 i.v., Melan-A VLP (★, *n* = 10); αCTLA-4 i.d., αPD-1 i.v., Melan-A VLP (◆, purple, *n* = 10); and αCTLA-4 i.d., αPD-1 i.d., Melan-A VLP (⬣, orange *n* = 15). (**B**) Systemic delivery of αPD-1 in unvaccinated animals dosed with αCTLA-4 further decreases primary tumor growth. Non-treatment (▽, *n* = 5); αCTLA-4 i.d., αPD-1 i.v., (▼, blue *n* = 5); αCTLA-4 i.v., αPD-1 i.v., (■, *n* = 10); and αCTLA-4 i.d., αPD-1 i.d., (△, pink, *n* = 15). (**C**) Lymphatic delivery of αCTLA-4 in combination with αPD-1 increases CD45^+^CD3^+^ T-cells in the tumor microenvironment. (**D**) Lymphatic delivery of αCTLA-4 in combination with αPD-1 in vaccinated animals changes the distribution of CD4^+^ and CD8^+^ in the tumor microenvironment. Non-treatment (▽, *n* = 3); αCTLA-4 i.v., αPD-1 i.v., (■, *n* = 5); αCTLA-4 i.d., αPD-1 i.v., (▼, blue, *n* = 3); αCTLA-4 i.d., αPD-1 i.d., (△, pink, *n* = 5).; αCTLA-4 i.v., αPD-1 i.v., Melan-A VLP (★, *n* = 5); αCTLA-4 i.d., αPD-1 i.v., Melan-A VLP (◆, purple, *n* = 5); αCTLA-4 i.d., αPD-1 i.d., Melan-A VLP (⬣, orange *n* = 6). *, ** Denotes statistical significance by unpaired, two-tailed Student’s *t* test (*p* < 0.05, 0.005). Error bars indicate SEM. i.d., intradermal; i.v., intravenous.

**Figure 8 cancers-14-01823-f008:**
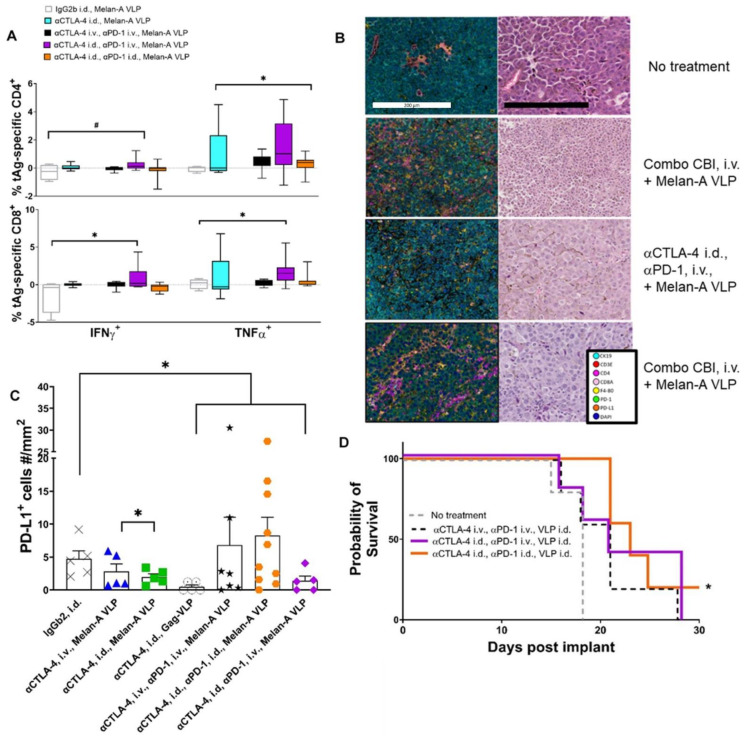
Impact of anti-tumor immunity on tumor microenvironment (TME) and survival. (**A**) Changes in Melan-A TNFα and IFNγ functionality of CD8^+^ and CD4^+^ cells in PBMCs. Isotype IgG2b, Melan-A VLP (open bar, *n* = 4); αCTLA-4 i.d., Melan-A VLP (blue, *n* = 9); αCTLA-4 i.v., αPD-1 i.v., Melan-A VLP (black, *n* = 7); αCTLA-4 i.d., αPD-1 i.v., Melan-A VLP (purple, *n* = 10); and αCTLA-4 i.d., αPD-1 i.d., Melan-A VLP (orange *n* = 10). *, ^#^ Denotes statistical significance by unpaired, two-tailed and one-tailed Student’s *t* test (*p* < 0.05), respectively. Error bars indicate SEM. (**B**) Multiplex immunofluorescence (mIF) and H&E of tumor tissue harvested at 17 days post implant (d.p.i.) in (i) untreated animals and animals treated with (ii) αCTLA-4 i.v., αPD-1 i.v., Melan-A VLP; (iii) αCTLA-4 i.d., αPD-1 i.d., Melan-A VLP; and (iv) αCTLA-4 i.d., αPD-1 i.v., Melan-A VLP. Scalebars in miF and H&E panels correspond to 200 μm. (**C**) Multiplex immunofluorescence of PD-L1^+^ cells (CK^+^, F40/80^+^ and CD45^+^CD3E^+^) in TME at 17 d.p.i. for animals treated with IgG2b i.d. (X, grey, *n* = 5); αCTLA-4 i.v., Melan-A VLP (▲, blue, *n* = 5); αCTLA-4 i.d., Melan-A VLP (■, green, *n* = 5); αCTLA-4 i.d., Gag-VLP (○, *n* = 5); αCTLA-4 i.v., αPD-1 i.v., Melan-A VLP (black, *n* = 5); αCTLA-4 i.d., αPD-1 i.d., Melan-A VLP (⬣, orange *n* = 6); and αCTLA-4 i.d., αPD-1 i.v., Melan-A VLP (◆, purple *n* = 5). * Denotes statistical significance by unpaired, two-tailed Student’s *t* test (*p* < 0.05). Error bars indicate SEM. (**D**) Kaplan-Meier survival curve shows lymphatic delivery of combinational therapy with Melan-A VLP significantly improves survival up to 30 d.p.i. * Indicates significant difference between αCTLA-4 i.d., αPD-1 i.d., Melan-A VLP, and no treatment. i.d., intradermal; i.v., intravenous. Scale bar: 200 μm.

**Figure 9 cancers-14-01823-f009:**
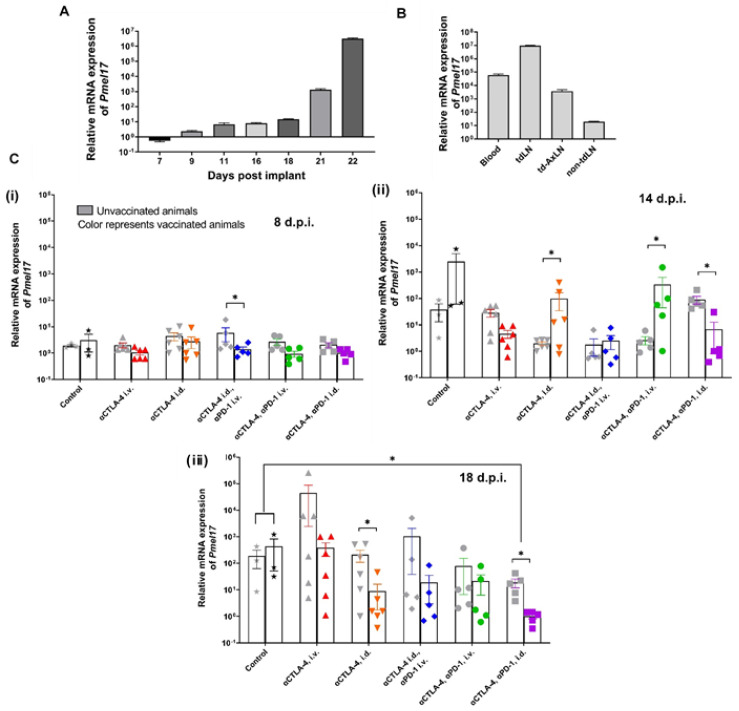
Metastatic burden in a dorsal foot implant model and response to CBIs in vaccinated and unvaccinated animals. (**A**) Transcriptomic levels of *Pmel17* in blood increases during tumor progression. (**B**) Transcriptomic levels of melanoma marker *Pmel17* in blood and LN at 11 days post implant (d.p.i.) coincides with visible metastasis (*n* = 5). (**C**) Relative mRNA expression of *Pmel17* in blood at (i) 8 d.p.i., (ii) 11 d.p.i., and (iii) 18 d.p.i. for vaccinated (denoted by colored symbols) and unvaccinated animals (denoted by grey symbols). * Denotes statistical significance by unpaired, two-tailed Student’s *t* test (*p* < 0.05). Error bars indicate SEM. tdLN, tumor draining lymph node; td-AxLN, tumor draining axillary lymph node; non-tumor draining lymph node (non-tdLN); i.d. intradermal; i.v. intravenous.

## Data Availability

Data was generated by the authors and is available upon request.

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
