# Peer review of "Enhanced T-Cell Priming and Improved Anti-Tumor Immunity through Lymphatic Delivery of Checkpoint Blockade Immunotherapy"

_cancers, 2022, doi:10.3390/cancers14071823_

Round 1
Reviewer 1 Report
The paper my Carolina Mantilla-Rojas et al describes that the combined vaccination and checkpoint therapy through lymphatic delivery is very efficient in a mouse model of melanoma (B16F10). The author are experts in the lymphatic delivery technique for immunotherapy. Thus, the presented results are quite novel. The general conclusions of the paper are supported by the results. However, the mechanistic explanation of the results is in some instances very speculative.
This is due to the use of a limited number of markers to analyze the TILs.
The quality of the paper could be improved by using other markers that can define subsets such as Treg or exhausted CD8 T cells? These data could increase the interest and interpretation of the obtained results.
The quality of the figures has to be improved. Most of the figures are too small, especially Figure 1 C and D; Figure 2; Figure 5, Figure 6 and Figure 7. This is mandatory to properly see the presented results.
Author Response
The paper my Carolina Mantilla-Rojas et al describes that the combined vaccination and checkpoint therapy through lymphatic delivery is very efficient in a mouse model of melanoma (B16F10). The author are experts in the lymphatic delivery technique for immunotherapy. Thus, the presented results are quite novel. The general conclusions of the paper are supported by the results. However, the mechanistic explanation of the results is in some instances very speculative.
This is due to the use of a limited number of markers to analyze the TILs.
We thank the reviewer for the complementary comments. We strongly agree that the mechanistic explanation of the results is limited by the ability to characterize TILs as well as other the cell types in the TME. Because the limited number of markers inherently restrict FACS and mIF analyses, we are currently conducting single cell RNA sequencing to offer the maximal information to decipher the mechanisms by which LNs can mediate the tumor immune microenvironment. In response to the reviewer’s comment, we revise our sentences in the Discussion regarding future work:
FACS evaluation of Melan-A functionality of TILs and of other immune cell types in the TME was limited by cell numbers. Future work to evaluate how lymphatic delivery and VLP vaccination change the TCR repertoire and transcriptomic phenotypes of TILs using techniques such as single cell RNA sequencing may better demonstrate the mechanisms behind how pharmacological manipulation of dLNs can impact the immune status of the TME.
The quality of the figures has to be improved. Most of the figures are too small, especially Figure 1 C and D; Figure 2; Figure 5, Figure 6 and Figure 7. This is mandatory to properly see the presented results,.
We have modified all the Figures in the contribution.
Reviewer 2 Report
Reviewer comments:
Cancer immunotherapy has made impressive effect in the last decade. This study focused on the checkpoint blockade immunotherapies and utilization of virus like particles, which is a very relevant therapy because these medications can help the body's immune system recognize and attack cancerous cells. The authors have investigated a very relevant topic in terms of utilising checkpoint blockade immunotherapies in which lymphatic delivery of CBI (αCTLA-4 and αPD-1) to tumor dLNs (tdLNs) improves anti-tumor responses over intravenous administration and that vaccination against tumor associated antigen further enhances these responses.
This study brings novelty to the field to treat cancer by applying combinatorial strategies of virus like particle (VLP) vaccination against tumor associated antigen with checkpoint blockade inhibitor, the anti-tumor responses were significantly robust and enhanced survival benefits.
Their findings shows that lymphatic delivery of CBIs reduced primary tumor size and metastatic tumor burden, alleviated the pro-tumorigenic immune environment, and improved survival over systemic administration of CBIs.
This manuscript is for the most part well written with substantial evidence of confirmatory data. The experimental designing is impressive, and the data provided was comprehensive. The discussion is also well goes with the results and postulated according to the evidence provided. The references are appropriate and timely.
Minor criticisms
• Figures quality needs to be improved.
• Please undergo a thorough check of the manuscript for typographical and grammatical errors.
Author Response
Cancer immunotherapy has made impressive effect in the last decade. This study focused on the checkpoint blockade immunotherapies and utilization of virus like particles, which is a very relevant therapy because these medications can help the body's immune system recognize and attack cancerous cells. The authors have investigated a very relevant topic in terms of utilising checkpoint blockade immunotherapies in which lymphatic delivery of CBI (αCTLA-4 and αPD-1) to tumor dLNs (tdLNs) improves anti-tumor responses over intravenous administration and that vaccination against tumor associated antigen further enhances these responses.
This study brings novelty to the field to treat cancer by applying combinatorial strategies of virus like particle (VLP) vaccination against tumor associated antigen with checkpoint blockade inhibitor, the anti-tumor responses were significantly robust and enhanced survival benefits.
Their findings shows that lymphatic delivery of CBIs reduced primary tumor size and metastatic tumor burden, alleviated the pro-tumorigenic immune environment, and improved survival over systemic administration of CBIs.
This manuscript is for the most part well written with substantial evidence of confirmatory data. The experimental designing is impressive, and the data provided was comprehensive. The discussion is also well goes with the results and postulated according to the evidence provided. The references are appropriate and timely.
We thank the reviewer for the complementary comments.
Minor criticisms
Figures quality needs to be improved.
We have revised all the figures in the manuscript.
Please undergo a thorough check of the manuscript for typographical and grammatical errors.
We have secured an independent reviewer and have corrected typographical and grammatical errors. We have made edits to the language to improve clarity.
Reviewer 3 Report
In present research, authors uses murine model to show that antigen specific T cell priming delivered through vaccination and immunotherapy helps to improve anti-tumor responses. Authors here advocate the lymphatic delivery to be superior over normal i.v. route and reflected in better prognosis in animals, I have several reservations, my comments are appended a below:
Major concerns:
- My primary concern is that how the proposed route of administration is clinically relevant? It may not be feasible to deliver does through lymphatic delivery. Authors should comment on this.
- Animal experiments: do author perform toxicological analysis (body weight, serum AST measurement) at end point?
- Figure 3- introduce scheme of experiment. Doses of drugs should be indicated. Authors should share HE images.
- In figure 3- do authors check other immune cells as MDSC/ Tregs?
5.Figure 4- authors should supplement with FACS raw data.
- Melan A looks dominant antigen while in clinical stetting, the tumor are often heterogeneous. How do authors coincide this?
- Figure 5 D-, figure 6 A, B annotate with statistical inference.
- Authors should include hypothesis figure for better understanding.
Minor concerns:
- Introduction- authors should share details on prognosis observed in clinic on selected immunotherapies.
- Authors should note catalogue no of all reagents/kits used.
- Figure legend- indicate the type of statistical test used.
- immunotherapy is also know to be affected by other cofounders. For instance, authors may refer PMID: 33076303 and add few lines.
Author Response
In present research, authors uses murine model to show that antigen specific T cell priming delivered through vaccination and immunotherapy helps to improve anti-tumor responses. Authors here advocate the lymphatic delivery to be superior over normal i.v. route and reflected in better prognosis in animals, I have several reservations, my comments are appended a below:
Major concerns:
- My primary concern is that how the proposed route of administration is clinically relevant? It may not be feasible to deliver does through lymphatic delivery. Authors should comment on this.
While we have addressed this in the final sentences in the discussion section, in response to the reviewer’s comment, we expand upon this further by adding references and providing an alternative:
Finally, it is important to note that i.d. administration volumes through conventional hypodermic needles are clinically limited to 100 µL or less, potentially limiting the direct delivery CBI to dLNs. Future development of microneedle arrays for accurate intradermal infusion of clinically relevant amounts and/or volumes of CBI is underway by several academic laboratories and drug delivery companies [47, 48]. Alternatively, s.c. administration or implantation of s.c., drug-eluding devices within an intact lymphatic watershed could also reach LNs, although with less bioavailability than i.d. administration.
- Animal experiments: do author perform toxicological analysis (body weight, serum AST measurement) at end point?
We did not perform toxicity analyses because WT mice do not provide a good model for recapitulating irAEs. We are currently conducting these same measurements on a transgenic model with transient knock-out of Tregs. Because this transgenic model is likely to overpredict anti-tumor responses, it was not appropriate to use an irAEs model to show that lymphatic delivery improves anti-tumor responses. Nonetheless, in response to the reviewer’s comment we expand the following section in the Discussion:
If tAg specific T cell priming and maintenance is proven to be the initiating, primary driver of therapeutic efficacy, then CBI doses could be reduced to maximize exposure to dLNs and minimize systemic exposure, thereby reducing tAg-indiscriminate immune responses and potentially reducing severe immune-related adverse toxicities. Further study is needed to determine whether regional delivery to tdLNs can reduce irAEs by limiting systemic exposure. Because wild-type mice do not provide reliable readouts that can indicate clinical irAEs, future studies are needed in transgenic models of transient Treg depletion [46] to show reduction in antigen indiscriminate T cell expansion and infiltrating lymphocytes in normal tissues with regional, lymphatic as opposed to systemic delivery of CBI. Because Treg depletion will likely enhance responses to CBI, these studies are best performed in non-tumor bearing animals.
- Figure 3- introduce scheme of experiment. Doses of drugs should be indicated. Authors should share HE images.
The scheme of the experiment is described in modified Figure 2 and in response to the reviewer’s comment, the doses are now included in the figure. All miF images are now accompanied by H&E in both the main text and in the supplemental section.
- In figure 3- do authors check other immune cells as MDSC/ Tregs?
This comment is identical to that from Reviewer #1. In response to both Reviewer #1 and #3 we amend the discussion with the limitation and future work needed. In addition, we point out that the validated mIF panel expands the immune cells interrogated in the TME, but still is limited.
Future work to evaluate how lymphatic delivery and VLP vaccination change the TCR repertoire and transcriptomic phenotypes of TILs using techniques such as single cell RNA sequencing may better demonstrate the mechanisms behind how pharmacological manipulation of dLNs can impact the immune status of the TME.
- Figure 4- authors should supplement with FACS raw data.
In response to the reviewer’s comment, we add the FACS data for former Figure 4.
- Melan A looks dominant antigen while in clinical stetting, the tumor are often heterogeneous. How do authors coincide this?
We are not certain we understand the reviewer’s abbreviated comment, but anticipate that Melan-A is a TAA which may be susceptible to tolerization by several mechanisms, including by loss of tumor MHC alleles, and elimination by central tolerance mechanism. In response to the reviewer’s comment, we further expand the Discussion and include:
It is important to note that the Melan-A/Mart antigen used herein is a TAA. As a result, Melan-A-specific T cells can be subject to elimination by central tolerance mechanisms thereby limiting efficacy and resulting in heterogenous responses. In addition, allelic MHC I loss by tumor cells under immune pressure can result in futile T-cell priming and additional mechanisms of tumor immune escape [42]. Multi-target vaccination and selection of multiple tumor neoantigens could limit tumor immune escape [23, 37].
- Figure 5 D-, figure 6 A, B annotate with statistical inference.
In response to the reviewer’s comment, we have re-evaluated all data in the contribution and better represented the statistical differences.
- Authors should include hypothesis figure for better understanding.
In response to the reviewer’s comment, we move the Graphical Abstract into the body of the text as agreed upon by the Editorial staff and renumber the Figures.
Minor concerns:
- Introduction- authors should share details on prognosis observed in clinic on selected immunotherapies.
In response we add the following into the Introduction:
Because aCTLA-4 monotherapy is associated with lower response rates and higher rates of Grade 3-4 toxicities than aPD-1 monotherapy [1-3], aPD-1 monotherapy has become the preferred immunotherapy in patients with advanced melanoma [4, 5]. For those patients unresponsive to monotherapy, combination of aCTLA-4 and aPD1 therapies have been shown to have complementary activity of up to 50-60% response rates in advanced Stage III or IV melanoma, but disappointingly, they act synergistically to amplify immune-related adverse events (irAEs) and severe toxicity in up to 60% of all patients [2, 3, 6-9]. Sequential monotherapies of anti-PD1 followed by anti-CTLA-4 or the reverse sequence have optimal response rates of 30-40% in advanced metastatic melanoma, but again with toxicity rates of ~50% [10].
- Authors should note catalogue no of all reagents/kits used.
We have added the Catalogue number of all reagents/kits used.
- Figure legend- indicate the type of statistical test used.
We have listed the statistical tests in the text and now in the legends.
- immunotherapy is also know to be affected by other cofounders. For instance, authors may refer PMID: 33076303 and add few lines.
In response to the reviewer’s comments we add:
While confounders of CBI include diet, age, gender, lifestyle, and chronic disorders [11], a focused approach to improve tumor-specific, and alleviate tumor-irrelevant, immune responses could improve response rates.
Round 2
Reviewer 3 Report
All my comments are answered.